# 5-Chloro-2-Guanidinobenzimidazole (ClGBI) Is a Non-Selective Inhibitor of the Human H_V_1 Channel

**DOI:** 10.3390/ph16050656

**Published:** 2023-04-27

**Authors:** Tibor G. Szanto, Adam Feher, Eva Korpos, Adrienn Gyöngyösi, Judit Kállai, Beáta Mészáros, Krisztian Ovari, Árpád Lányi, Gyorgy Panyi, Zoltan Varga

**Affiliations:** 1Department of Biophysics & Cell Biology, Faculty of Medicine, University of Debrecen, 4032 Debrecen, Hungary; szanto.gabor@med.unideb.hu (T.G.S.); feher.adam@med.unideb.hu (A.F.); korpos.eva@med.unideb.hu (E.K.); meszaros.beata@med.unideb.hu (B.M.); krisztianovari99@gmail.com (K.O.); panyi@med.unideb.hu (G.P.); 2ELKH-DE Cell Biology and Signaling Research Group, Faculty of Medicine, University of Debrecen, 4032 Debrecen, Hungary; kallai.judit@med.unideb.hu; 3Department of Immunology, Faculty of Medicine, University of Debrecen, 4032 Debrecen, Hungary; gadrienn@med.unideb.hu (A.G.); alanyi@med.unideb.hu (Á.L.)

**Keywords:** ClGBI, proton channel, voltage-gated ion channels, blocking kinetics, selectivity screening

## Abstract

5-chloro-2-guanidinobenzimidazole (ClGBI), a small-molecule guanidine derivative, is a known effective inhibitor of the voltage-gated proton (H^+^) channel (H_V_1, *K_d_* ≈ 26 μM) and is widely used both in ion channel research and functional biological assays. However, a comprehensive study of its ion channel selectivity determined by electrophysiological methods has not been published yet. The lack of selectivity may lead to incorrect conclusions regarding the role of hHv1 in physiological or pathophysiological responses in vitro and in vivo. We have found that ClGBI inhibits the proliferation of lymphocytes, which absolutely requires the functioning of the K_V_1.3 channel. We, therefore, tested ClGBI directly on hK_V_1.3 using a whole-cell patch clamp and found an inhibitory effect similar in magnitude to that seen on hH_V_1 (*K_d_* ≈ 72 μM). We then further investigated ClGBI selectivity on the hK_V_1.1, hK_V_1.4-IR, hK_V_1.5, hK_V_10.1, hK_V_11.1, hK_Ca_3.1, hNa_V_1.4, and hNa_V_1.5 channels. Our results show that, besides H_V_1 and K_V_1.3, all other off-target channels were inhibited by ClGBI, with *K_d_* values ranging from 12 to 894 μM. Based on our comprehensive data, ClGBI has to be considered a non-selective hH_V_1 inhibitor; thus, experiments aiming at elucidating the significance of these channels in physiological responses have to be carefully evaluated.

## 1. Introduction

The voltage-gated proton channel (H_V_1) is a relatively recently identified ion channel that considerably differs from other voltage-gated ion channels due to the lack of a pore domain [1]. Accordingly, the voltage-sensing domain (VSD) of H_V_1 has the dual role of voltage sensing and establishing proton permeation. The crystal structure of mouse H_V_1 has been presented in the resting state, providing a detailed view for understanding the general principles of voltage sensing and proton permeation [2]. H_V_1 is expressed in a wide variety of tissues and consequently has been linked to various cellular functions, such as pH regulation [3,4], proliferation [5], migration [6,7], and reactive oxygen species (ROS) production [8]. It was also found that H_V_1-deficient B cells had impaired antibody responses in vivo; thus, H_V_1 plays a role in the humoral immune response, as well [9]. In addition, H_V_1 has been implicated in the development of diseases related to excessive ROS production [9,10] and various tumors [5,9,11,12,13,14]. These findings suggest a potential therapeutic use of H_V_1-modulating compounds. Consequently, a number of studies have emerged for finding prospective proton channel inhibitors [15,16,17]. One of the first families of H_V_1 inhibitors was benzimidazoles, which are often prescribed as proton pump inhibitors [18], and their guanidine derivatives [19]. Multiple derivatives of the molecule 2-guanidinobenzimidazole (2GBI) were found to effectively block H_V_1 from the intracellular side; however, it was 5-chloro-2-guanidinobenzimidazole (ClGBI) that raised the greatest interest as a research tool due to its highest affinity and much-improved ability to cross the membrane, thereby offering the possibility of extracellular applications [20]. Although several other blockers of H_V_1 have been reported since then [15,16,17,21], ClGBI seems to remain the most frequently used tool to suppress H_V_1 currents.

Recently, the expression of H_V_1 channels in myeloid-derived suppressor cells (MDSCs) and their essential role in the inhibition of T cell proliferation by MDSCs have been reported [22]. Pretreatment of MDSCs with ClGBI reduced their suppressive effect on T cell proliferation, suggesting the potential therapeutic use of H_V_1 inhibitors in immunoregulation. However, the presence and possible functional roles of voltage-gated proton currents have also been described in Jurkat T cells, as well as murine and human peripheral T cells [23] and recently in activated T cells [24]. Thus, the application of ClGBI in the presence of both MDSCs and T cells may also directly affect the function of the latter cell type, as well. We, therefore, tested the effect of ClGBI treatment on the proliferation of peripheral blood lymphocytes, the majority of which are T cells, and found a significant suppression by a 200 μM concentration of the drug. As the K_V_1.3 voltage-gated potassium channel is a crucial regulator of the membrane potential both in quiescent and activated/proliferating T cells [25], we hypothesized that the effect of ClGBI may occur at least in part via K_V_1.3 inhibition. Therefore, we first assessed its effect directly on hK_V_1.3 using electrophysiological methods and observed current inhibition in a concentration range comparable to the inhibition of H_V_1. Although ClGBI is widely used in the field of ion channel research as a potent H_V_1 blocker [7,17,22], a comprehensive study of its selectivity with electrophysiological methods has not been published yet. Since the suppression of proliferation and K_V_1.3 blockade suggested that ClGBI may be a non-selective inhibitor, it motivated us to conduct further patch clamp experiments to perform a selectivity screening of ClGBI beyond K_V_1.3. Measurements were carried out on hK_V_1.1, hK_V_1.4-IR, hK_V_1.5, hK_V_10.1, hK_V_11.1 (hERG), hK_Ca_3.1, hNa_V_1.4, and hNa_V_1.5. ion channels. All the tested channels were inhibited to some extent by 200 μM ClGBI, so our results clearly indicate that ClGBI is not a highly selective inhibitor of H_V_1 channels, as had been assumed in several previous studies.

## 2. Results

### 2.1. ClGBI Inhibits the Human H_V_1 Channel Expressed in CHO Cells

It has been shown that the 2-guanidinobenzimidazole (2GBI) derivative ClGBI inhibits H_V_1 proton channels expressed in *Xenopus laevis* oocytes by binding to the VSD intracellularly as well as by blocking when applied extracellularly due to its ability to cross the membrane [20]. First, we aimed to confirm the effects of ClGBI on the hH_V_1 channel in our system. The macroscopic currents were measured in transiently transfected CHO cells (see Section 4 for details). Figure 1A shows representative whole-cell current traces evoked by 1 s voltage ramps to +60 mV from a holding potential of –60 mV with an intersweep interval of 10 s. Voltage ramps allow the simultaneous measurement of current amplitude at a given membrane potential and activation threshold voltage of the channels. ClGBI was applied to the bath solution, and the amplitudes of the currents were measured at the end of the 1-s-long voltage pulses. A robust inhibition was observed: at the equilibrium block, 200 μM ClGBI caused a major reduction in current amplitude, and the remaining current fraction (*RCF*) was 0.13 ± 0.01, n = 7 (Figure 1A, red). The proper operation of the perfusion apparatus was confirmed using ECS at a pH of 6.4, which significantly shifts the opening threshold of the channel in the positive direction as the gating of H_V_1 depends on the transmembrane pH gradient.

The time course of the inhibition of H_V_1 currents at 200 μM ClGBI is shown in Figure 1B. Normalized peak currents as *RCF* were plotted as a function of time. The equilibrium block was reached in ~100 s at this concentration. In contrast to the relatively fast association kinetics of ClGBI, the dissociation rate was extremely slow; accordingly, recovery up to ~10% of the control current took several minutes. This observation is in agreement with the slow washout kinetics also seen in outside-out patch recordings [20].

Figure 1C shows the concentration–response experiments performed for testing the concentration-dependent inhibition of H_V_1 by ClGBI. Different concentrations of ClGBI were applied to the cells for an adequate duration to reach the equilibrium block, considering the slower blocking kinetics at lower ClGBI concentrations. The *RCF*s were calculated at each ClGBI concentration and plotted as a function of ClGBI. Fitting the data points yielded *K_d_* = 15.9 ± 2.0 µM and an *n_H_* of 1.0 ± 0.1 (n = 4).

### 2.2. ClGBI Also Inhibits the K_V_1.3 Channel of Lymphocytes

Recently, the increased expression of H_V_1 has been shown in activated T cells, suggesting a link to proliferation [24]. We have, therefore, investigated the effect of ClGBI on lymphocyte proliferation induced by PHA activation using a CFSE assay. In this method, following the loading of cells with CFSE dye, each round of cell division is apparent by the reduced dye content and consequent lower fluorescence intensity of the daughter cells. Successful activation and the resulting proliferation of the cells were demonstrated by the shift of the fluorescence histogram toward lower values and the appearance of multiple peaks (Figure 2). The application of 200 μM of ClGBI completely blocked the proliferation of the cells, as shown by the single peak fluorescence histogram in the presence of the drug overlapping with the main peak of the non-activated cells. The inhibitory effect on proliferation was much weaker in the presence of 20 μM ClGBI and completely absent at 2 μM (Figure 2).

The essential role of voltage-gated K_V_1.3 channels in T cell activation and proliferation is well-documented [26,27,28], so we tested whether the proliferation was suppressed in part via a K_V_1.3 blockade. Although we did not separate T cells, they make up most of the mixed lymphocyte population, and the proliferation of B cells also relies on the activity of the K_V_1.3 potassium channel [29]. The inhibitory effect of ClGBI on these channels was assessed by patch clamp. hK_V_1.3 currents were evoked in human lymphocytes by a series of 15-ms-long depolarization pulses to +50 mV from a holding potential of −120 mV (Figure 3A). The open probability of the channel at +50 mV is maximal, and the relatively short duration of the depolarizing periods prevented inactivation. The time between voltage pulses was set to 15 s to avoid the cumulative inactivation of hK_V_1.3 channels. Under the given experimental conditions (i.e., the voltage protocol used and the lack of Ca^2+^ in the pipette solution), the whole-cell currents were conducted exclusively by hK_V_1.3 channels. Figure 3A shows macroscopic K^+^ currents through hK_V_1.3 channels were recorded sequentially in the same cell in the absence (control, black) and presence of 200 μM ClGBI (red) dissolved freshly in the ECS. At the equilibrium block, 200 μM ClGBI caused a ~75% reduction in current amplitude (*RCF* was 0.25 ± 0.01, n = 9). ECS containing 10 mM tetraethylammonium (TEA^+^) was used to verify the identity of the ion channel and the proper operation of the perfusion system (10 mM TEA^+^, blue).

The kinetics of the development of the inhibition of K_V_1.3 current by 200 μM ClGBI is shown in Figure 3B. The K_V_1.3 current is progressively blocked by ClGBI, demonstrated by the decrease in the normalized peak currents as a function of time, and the origin (t = 0) corresponds to the start of the perfusion with ClGBI dissolved in ECS. The kinetics of the block followed a single exponential time course, indicating a simple bimolecular interaction between ClGBI and the channel yielding τ_on_ = 22.1 ± 2.7 s (n = 3). Figure 3B also shows that the inhibition of the K_V_1.3 current is reversible; after the equilibration block was reached, the perfusion was switched to control ECS (washout), resulting in complete recovery from current inhibition.

We also performed a concentration–response experiment series for hK_V_1.3 channels by ClGBI. Different concentrations of ClGBI were added to the cells until a complete equilibrium block was reached and *RCF* values were determined. Data points on the dose–response curve represent the mean of 4–5 individual measurements and are fitted with a two-parameter inhibitor vs. response model (see Materials and Methods for details) to obtain the dissociation constant (*K_d_*) and Hill coefficient (*n_H_*), which were *K_d_* = 72.4 ± 4.0 µM and *n_H_* = 1.1 ± 0.07, respectively. The comparison of Figure 1B and Figure 3B clearly indicates that although ClGBI has a slightly higher affinity for H_V_1 than K_V_1.3, the two channels are inhibited in the same concentration range. Therefore, when ClGBI is applied in systems expressing both channels, their simultaneous inhibition must be considered.

### 2.3. ClGBI Is not Selective for hHv1 as It Inhibits a Wide Range of Other Channels

Prompted by the discovery that K_V_1.3 is inhibited by ClGBI, we tested the effect of the compound on eight other channels, including different voltage-gated K^+^ and Na^+^ channels and the intermediate conductance Ca^2+^-activated K^+^ channel, to assess its selectivity. Whole-cell patch clamp currents recorded in the absence and presence of 200 μM ClGBI are shown in Figure 4 for hK_V_1.1 (Figure 4A), hK_V_1.4ΔN (fast inactivation-removed hK_V_1.4, Figure 4B), hK_V_1.5 (Figure 4C), K_V_10.1 (Figure 4D), hK_V_11.1 (Figure 4E), hK_Ca_3.1 (Figure 4F), hNa_V_1.4 (Figure 4G), and hNa_V_1.5 (Figure 4H). For voltage protocols and the composition of ECS and ICS, see Materials and Methods. The *RCF* values, measured at equilibrium inhibition, are summarized in Figure 4I. All the investigated channels were blocked significantly by ClGBI at 200 μM. Similar to K_V_1.3, the inhibition was almost fully reversible for most channels. Due to the low-affinity inhibition of the channels, a complete dose–response curve would have required very large quantities of ClGBI (and concomitant high DMSO concentrations); thus, we could only determine an estimated *K_d_* value for the investigated channels from a single concentration, assuming a bimolecular interaction between ClGBI and the ion channels. These were (in μM) 323.8 ± 13.1 for K_V_1.1 (n = 5), 188.4 ± 38 for K_V_1.4 (n = 6), 310.2 ± 43.6 for K_V_1.5 (n = 4), 77.5 ± 14.2 for K_V_10.1 (n = 5), 12.0 ± 2.1 for K_V_11.1 (n = 5), 893.9 ± 133.5 for K_Ca_3.1 (n = 5), 590.7 ± 122.7 for Na_V_1.4 (n = 4), and 186.7 ± 35.7 (n = 6) for Na_V_1.5.

## 3. Discussion

Although voltage-gated proton currents were first described more than three decades ago [30], the gene of H_V_1 was only identified in 2006 [1]. Since then, H_V_1 has been found in numerous species and a wide variety of cell types and associated with various cellular functions [3,6,31,32,33,34], of which its contribution to ROS production by immune cells may be the best known [10]. The excessive function or overexpression of H_V_1 has been linked to various disease conditions associated with pathologically excessive ROS generation and cancer development [35]. This potential involvement of H_V_1 in disease conditions promoted it to become a prospective drug target, which initiated a search for blocking compounds of high affinity and selectivity. Although several publications have reported the discovery of small molecule or peptide toxin blockers of H_V_1 [15,16,17,36], their effectiveness has not been confirmed in functional tests by other groups, so ClGBI remains the most generally used blocker for research purposes [22,37]. It has been used to identify the channel and demonstrate the functional role of H_V_1 in various biological systems suppressing viability, migration, and proliferation [7,37,38,39]. However, the ion channel selectivity of ClGBI has not been comprehensively verified, raising the possibility that in several studies, the observed effects occurred at least in part via the inhibition of other off-target ion channels.

For this reason, we aimed to screen the effect of ClGBI on several different ion channels using mammalian expression systems to obtain a reliable comparison. First, we tested the CHO cells for the presence of endogenous currents reported earlier [40] since these non-specific currents may interfere with the interpretation of pharmacological studies designed to characterize the effect of ClGBI on heterologously expressed ion channels. Thus, we recorded whole-cell outward currents evoked by short depolarizing pulses on native CHO cells; however, the measured current was negligible (on average, the peak amplitude was ~10 pA at +50 mV, data not shown). Thus, the overexpression of the heterologously expressed channels completely eliminates the contribution of the endogenous background current to the whole cell current by minimizing potential errors in the pharmacological data.

Guanidine derivatives, including ClGBI, were shown to inhibit H_V_1 activity in the μM range [19,20]. We confirmed these results using a mammalian expression system: the apparent *K_d_* and Hill coefficient for ClGBI were 15.9 ± 2.0 μM and 1.0 ± 0.1, respectively, which are in good agreement with the previously determined parameters measured using an amphibian expression system [20]. The slight difference in the observed *K_d_* values may be due to the different expression systems and/or the different availability of possible interacting partners of the channel in CHO cells and *Xenopus laevis* oocytes. Due to the relatively low affinity of ClGBI for H_V_1, higher concentrations of the compound would have required DMSO concentrations high enough to risk possible permeabilization of the membrane or other non-specific solvent effects.

In vivo studies of K_V_1.3 blocker peptide toxins have clearly shown their potential in suppressing T-cell-mediated inflammatory reactions, and K_V_1.3 inhibitors, in general, have long been known to suppress T cell activation and proliferation [26,27]. When performing functional H_V_1 assays on peripheral lymphocytes (mostly T cells), we observed progressively reduced proliferation after 6 days of treating the cells with increasing ClGBI concentrations, which led us to the hypothesis that ClGBI inhibits K_V_1.3 as well. We directly tested this by the patch clamp, and 200 μM ClGBI blocked ~80% of the K_V_1.3 current in a reversible manner, confirming the comparable affinities of ClGBI for H_V_1 and K_V_1.3. The current block developed at comparable rates for H_V_1 and K_V_1.3, requiring 50–100 s to reach block saturation at 200 μM, implying similar association rates (Figure 1B and Figure 3B). A major rate-limiting factor may be diffusion through the membrane, as the ClGBI block developed much more rapidly in inside-out patches than outside-out patches [20]. However, dissociation of ClGBI was significantly slower from H_V_1 than K_V_1.3, suggesting different interactions of the drug with the two channels. Structurally, K_V_1.3 has four VSDs, similar to H_V_1, but it also differs in that it has a “real” pore domain responsible for ion permeation, so the binding site of ClGBI on the two channels may not necessarily be at homologous locations [41,42].

Due to high structural similarity, small-molecule blockers often have similar affinities for different members within the same ion channel family, such as tetrodotoxin (TTX) blocking Na_V_ channels or tetraethylammonium blocking K^+^ channels [43]. This prompted us to test the effect of ClGBI on other members of the Shaker (K_V_1.x) family and then expand the assessment to the K_V_ channels of other families and then even further to a few non-K_V_ channels. As K_V_1 channels share high sequence homology and tend to form functional heterotetrameric structures in different tissues, we first tested the effect of 200 μM of ClGBI on channels closely related to K_V_1.3: K_V_1.1, K_V_1.4-IR, and K_V_1.5. For K_V_1 channels, the voltage sensor of a given subunit couples to the pore domain of the adjacent subunit, leading to domain-swapped architecture [44,45,46]. In contrast, for members of the K_V_10 (EAG) and K_V_11 (ERG) families [47,48], as opposed to the “classic” arrangement, the voltage sensor is not domain-swapped, i.e., the voltage sensor of a given subunit is coupled to the pore domain of the same subunit. Since the non-swapped voltage sensors should work to transmit force and regulate the gate in a way that is different from the lever mechanism proposed for Shaker-like K_V_ channels, we decided to test the effect of ClGBI on K_V_10.1 and K_V_11.1 channels, as well. Our patch clamp experiments revealed that all these K_V_ off-target channels were also inhibited by 200 µM of ClGBI without major changes to gating kinetics.

We also tested the effect of ClGBI on the intermediate-conductance calcium-activated K^+^ channel, K_Ca_3.1 (also known as IK_Ca_1 or SK_4_), which has high biological relevance, as it regulates membrane potential and calcium signaling in a wide variety of cell types, such as erythrocytes, activated T and B cells, macrophages, microglia, vascular endothelial cells, epithelial cells, and vascular smooth muscle cells [49,50]. Consequently, K_Ca_3.1 is suggested as a potential therapeutic target in diseases, such as anemia, atherosclerosis, and autoimmunity. K_Ca_3.1 differs from the other off-target channels since it is only gated by intracellular Ca^2+^ and lacks functional VSDs. Based on this, we considered K_Ca_3.1 worth including as a potential off-target channel, as well. Although 200 μM ClGBI inhibited K_Ca_3.1 to a small extent, its affinity for the channel was very low, most likely ruling out off-target effects in functional studies.

Voltage-gated sodium channels share a similar general structure with K_V_ channels containing four VSDs per channel. However, the functioning of these VSDs is less symmetrical than in K_V_ channels, and the pore domain is also different. This motivated us to investigate the effect of ClGBI on Na_V_ channels, which are molecular targets for a broad range of small molecules and peptides isolated from the venoms of scorpions, spiders, sea anemones, and cone snails with binding sites at varying locations on the VSDs or the pore domain [51]. We tested the effect of ClGBI on hNa_V_1.4 and hNa_V_1.5 channels, which are responsible for the generation and propagation of action potentials triggering muscle contraction in skeletal muscles and mediating the rising phase of the cardiac action potential, respectively [52]. They represent the two major classes of Na_V_ channels: Na_V_1.4 is blocked by low nanomolar concentrations of the guanidine-based neurotoxin TTX, a toxin isolated from puffer fish; thus, it is classified as a TTX-sensitive channel, whereas Na_V_1.5 is considered TTX-resistant, as it is inhibited by only high micromolar TTX concentrations. We have found that 200 μM ClGBI inhibited both channels, albeit with quite a low affinity, without altering gating kinetics. Its affinity was higher for the cardiac Na_V_1.5 channel, which should be considered when studying the role of H_V_1 in cardiac myocytes [38]. More importantly, the K_V_11.1 channel, crucial for the repolarization phase of the cardiac action potential, was blocked with approximately the same affinity as H_V_1.

In summary, we have tested the effect of ClGBI on the following ion channels: hK_V_1.1, hK_V_1.4-IR, hK_V_1.5, hK_V_10.1, hK_V_11.1, hK_Ca_3.1, hNa_V_1.4, and hNa_V_1.5. We have found a significant inhibitory effect of 200 μM ClGBI on all the investigated channels, although the *K_d_* values covered a wide concentration range (*K_d_* = 12–894 μM). The cardiac hERG channel, K_V_11.1, which is known to be blocked by numerous small molecules, thereby eliminating many of them as potential drug candidates, showed the highest affinity for ClGBI, equaling that of H_V_1. The affinity sequence was the following: H_V_1 » K_V_11.1 > K_V_1.3 » K_V_10.1 > K_V_1.4 » Na_V_1.5 > K_V_1.5 » K_V_1.1 > Na_V_1.4 > K_Ca_3.1.

For prospective ion channel inhibitor drug candidates, a molecule is generally expected to show at least 100 times greater affinity for the target channel than for available off-target channels. In functional studies, such as cell viability, proliferation, or migration, a similarly strict criterion would be preferable to clearly isolate the role of a given channel in the cellular function. The 200 μM ClGBI concentration used in our study blocked at least 20–25% of the current in all tested channels. Since this is identical to the concentration that was used in several functional assays [7,22], and most native cells express a wide variety of ion channels, the unequivocal identification of H_V_1 using ClGBI as the channel responsible for a given function is questionable.

Since H_V_1 lacks a pore domain, ClGBI must necessarily bind to the VSD. Yet, it blocks the proton permeation pathway within the VSD, similarly to classical pore blockers of other channels, rather than modifying VSD movement as gating modifiers do [20]. The residues responsible for binding have been identified by mutant cycle analysis. As the VSDs of K_V_ and Na_V_ channels do not normally have ion permeation pathways in them, it is unlikely that the ClGBI binding site is at a homologous location in these channels.

We have scanned the affinity of ClGBI on various voltage-gated ion channel targets (H_V_, K_V_, Na_V_) with various gating mechanisms (e.g., intracellular Ca^2+^ or voltage-gated channels) and various coupling mechanisms between the voltage sensor and the pore domain (e.g., domain-swapped and non-domain-swapped channels). Although we have not performed a detailed kinetic analysis of the currents, we observed no significant changes in gating kinetics upon ClGBI inhibition for any of the channels. This implies that ClGBI likely interacts with the pore domain of these ion channels by fully or partially plugging it rather than affecting VSD movement.

Our results clearly demonstrate that ClGBI inhibits a wide variety of ion channels with low affinity, some of them with an affinity comparable to H_V_1. Consequently, if any of these ion channels are suspected or proven to be expressed in the investigated cells or tissues, the use of ClGBI in such experiments as proof of H_V_1 function must be reconsidered. This fact underlines the great need for blockers of H_V_1 with much higher selectivity than ClGBI that could be used as more reliable research tools and potentially drug lead molecules.

## 4. Materials and Methods

### 4.1. Lymphocyte Proliferation

Peripheral blood mononuclear cells (PBMCs) were isolated from the heparinized blood of healthy donors (n = 3) using the Ficoll–Hypaque separation method. A CD14^−^ population obtained by negative selection with CD14 bead (cat. #130-050-201, Miltenyi Biotec B.V & CO. KG, Bergisch Gladbach, Germany)) PBMCs was incubated with carboxyfluorescein succinimidyl ester (CFSE; Sigma Aldrich/Merck KGaA, Darmstadt, Germany) at the final concentration of 1 µM in PBS for 15 min at 37 °C in the dark. After removing the unbound dye, 7.5 × 10^5^/mL cells were seeded in a 48-well plate in RPMI 1640 medium supplemented with 10% FBS, 1% GlutaMAX, and 1% penicillin–streptomycin and stimulated with phytohaemagglutinin (PHA, 5 μg/mL, Sigma Aldrich, St. Louis, MO, USA). The cells were grown in a humidified chamber at 37 °C and 5% CO_2_ for 6 days. As a negative control, unstimulated CFSE-labeled PBMCs were cultured in the same condition. ClGBI was added to the cells in concentrations of 200, 20 and 2 µM from day 0 of the proliferation. After 6 days of proliferation, the CFSE-labeled cells were collected and analyzed by flow cytometry in an ACEA NovoCyte 2000R cytometer (Agilent, Santa Clara, CA, USA).

### 4.2. Cells for Patch Clamp Recordings

Chinese hamster ovary (CHO) cells were cultured in Dulbecco’s modified Eagle’s medium (DMEM, Gibco, Thermo Fisher Scientific, Waltham, MA, USA, Cat# 11965084)) containing 10% fetal bovine serum (FBS, Sigma-Aldrich), 2 mM L-glutamine, 100 µg/mL streptomycin, and 100 U/mL penicillin-g (Sigma-Aldrich, St. Louis, MO, USA) at 37 °C in a 5% CO_2_ and 95% air humidified atmosphere. Cells were passaged twice per week following a 2–3 min incubation in PBS containing 0.2 g EDTA/L (Invitrogen, Waltham, MA, USA).

Human peripheral blood monocytes (PBMCs) were isolated from venous blood obtained from anonymized healthy donors. The peripheral blood mononuclear cells were isolated by Histopaque1077 (Sigma-Aldrich Hungary, Budapest, Hungary) density gradient centrifugation. The cells obtained were resuspended in Roswell Park Memorial Institute (RPMI) 1640 medium (Gibco, Cat# 11875085) containing 10% fetal calf serum (FCS, Sigma-Aldrich, St. Louis, MO, USA), 100 μg/mL penicillin, 100 μg/mL streptomycin, and 2 mM L-glutamine, seeded in a 24-well culture plate at a density of 5–6 × 10^5^ cells/mL, and grown in a 5% CO_2_ incubator at 37 °C for 3–5 days. Phytohemagglutinin A (PHA, Sigma-Aldrich, St. Louis, MO, USA) was also added at a concentration of 8–12 μg/mL to the medium to amplify the expression level of K_V_1.3. CHO cells and PBMCs were gently washed twice with 2 mL of ECS (see below) for the patch clamp experiments. hK_V_1.3 currents were recorded on activated lymphocytes 3 to 4 days after activation.

CHO cells were transiently transfected using a Lipofectamine 2000 kit (Invitrogen, Carlsbad, CA, USA) as per the manufacturer’s protocol with the following ion channel coding vectors: hHv1 (*hVCN1*, GenBank accession no. BC007277.1, a kind gift from Kenton Swartz, NIH, Bethesda, MD, USA), hK_V_1.1 (*hKCNA1* gene) in pCMV6-AC-GFP plasmid (Cat# RG211000, OriGene Technologies, Rockville, MD, USA), hK_V_1.4 (hK_V_1.4-IR, the inactivation ball deletion mutant of K_V_1.4; a kind gift from D. Fedida, University of British Columbia, Vancouver, BC, Canada), hK_V_1.5 in pEYFP plasmid (a kind gift from A. Felipe, University of Barcelona, Barcelona, Spain), hK_V_10.1 (hKCNH1 gene) in pCMV6-XL plasmid (Cat# SC303154, OriGene Technologies, Rockville, MD, USA), hK_Ca_3.1 (*hKCNN4* gene) in pEGFP-C1 vector (a kind gift from H. Wulff, University of California, Davis, CA, USA), and hNa_V_1.5 (*hSCN5A*, a kind gift from H. Abriel, University of Bern, Bern, Switzerland). The hH_V_1, hK_V_1.4-IR, and hK_V_10.1 channel plasmids were transiently co-transfected with a plasmid encoding the green fluorescent protein (GFP) at a molar ratio of 10:1, respectively. Transfected cells were washed twice with 2 mL of ECS (see below) and replated onto 35 mm polystyrene cell culture dishes (Cellstar, Greiner Bio-One, Kremsmünster, Austria). GFP-positive transfectants were identified with a Nikon Eclipse TS100 fluorescence microscope (Nikon, Tokyo, Japan) using bandpass filters of 455–495 nm and 515–555 nm for excitation and emission, respectively, and were used for current recordings (>70% success rate for co-transfection). Human embryonic kidney (HEK) 293 cells stably expressing hK_V_11.1 (*hERG*, *hKCNH2* gene, a kind gift from H. Wulff, University of California, Davis, CA, USA) and hNa_V_1.4 (*hSCN4A* gene, a kind gift from P. Lukács, Eötvös Loránd University, Budapest, Hungary) were also used. In general, ionic currents were recorded 24 to 36 h after transfection.

### 4.3. Electrophysiology

The standard whole-cell patch clamp method [53] was used to record ionic currents. Micropipettes were pulled in four stages using a Flaming Brown automatic pipette puller (Sutter Instruments, San Rafael, CA, USA) from GC 150F-15 borosilicate glass capillaries (Harvard Apparatus Co., Holliston, MA, USA) with tip diameters between 0.5 and 1 μm and heat-polished to a tip resistance ranging typically between 2 and 8 MΩ. All measurements were carried out using Axopatch 200B amplifiers connected to personal computers using Digidata 1550A data acquisition hardware (Molecular Devices Inc., Sunnyvale, CA, USA). In general, the holding potential was −120 mV. Records were discarded when a leak at the holding potential was more than 10% of the peak current at the given test potential. Experiments were conducted at room temperature, which ranged between 20 and 24 °C.

### 4.4. Solutions

For hK_V_1.1, hK_V_1.3, hK_V_1.4-IR, hK_V_1.5, hK_V_10.1, hNa_V_1.4, and hNa_V_1.5, the extracellular (bath) solution (ECS) contained 145 mM NaCl, 5 mM KCl, 2.5 mM CaCl_2_, 1 mM MgCl_2_, 10 mM Hepes, and 5.5 mM glucose (pH = 7.35 titrated with NaOH), and the intracellular (pipette) solution (ICS) contained 140 mM KF, 2 mM MgCl_2_, 1 mM CaCl_2_, 11 mM EGTA, and 10 mM Hepes (pH = 7.2 with KOH). For Na_V_ current recordings, Cs^+^-based ICS was used to avoid the recordings of endogenous K^+^ currents; thus, the ICS consisted of (in mM) 10 NaCl, 105 CsF, 10 HEPES, and 10 EGTA (pH = 7.2 titrated with CsOH). For hK_V_11.1, the ECS and ICS consisted of (in mM) 140 Choline-Cl, 5 KCl, 2 MgCl_2_, 2 CaCl_2_, 10 HEPES, 20 glucose, and 0.1 CdCl_2_ (pH = 7.35) and 140 KCl, 2 MgCl_2_, 10 HEPES, and 10 EGTA (pH = 7.3), respectively. For hK_Ca_3.1, the ECS contained 145 mM L-aspartic acid with Na, 5 mM KCl, 2.5 mM CaCl_2_, 1 mM MgCl_2_, 10 mM Hepes, and 5.5 mM glucose (pH = 7.4 with NaOH), and the ICS contained 145 mM K-Asp, 2 mM MgCl_2_, 8.5 mM CaCl_2_, 10 mM EGTA, and 10 mM Hepes (pH = 7.2 with KOH), giving ~2 µM free Ca^2+^ to fully activate the hK_Ca_3.1 current [54]. For hH_V_1, the ECS contained (in mM) 75 N-methyl D-glucamine (NMDG), 180 HEPES, 15 glucose, 3 MgCl_2_, and 1 EGTA (pH = 7.4 with CsOH), and the ICS contained (in mM) 75 NMDG, 180 MES, 3 MgCl_2_, 15 glucose, and 1 EGTA (pH = 6.4 with CsOH). The osmolarities of ECS and ICS were between 302 and 308 mOsm/L and ~295 mOsm/L, respectively.

Bath perfusion around the measured cell with different extracellular solutions was achieved using a gravity flow microperfusion system at a rate of 200 μL/min. Excess fluid was removed continuously. 5-chloro-2-guanidinobenzimidazole (ClGBI, (Sigma-Aldrich Hungary) solutions were made fresh in ECS from 100 mM stored at −20 °C. Stock solutions were prepared from powder dissolved in water-free DMSO (Sigma-Aldrich Hungary). ECS was supplemented with 0.2% DMSO. Positive controls were applied at a concentration equivalent to their *K_d_* values (0.3 mM and 10 mM TEA^+^ for K_V_1.1 and K_V_1.3, respectively, and 20 nM TRAM-34 for K_Ca_3.1). For hK_V_1.4, hK_V_1.5, and hK_V_11.1, high K^+^-based ECS was used as an indicator of the perfusion exchange, whose composition was identical to standard ECS except that it contained 150 mM KCl and 0 mM NaCl. For Na_V_ channels, a choline-based ECS was used as control, whose composition was (in mM) 145 choline-Cl, 5 KCl, 10 HEPES, 5.5 glucose, 2.5 CaCl_2_, and 1 MgCl_2_. For hH_V_1, ECS at pH 6.5 was used as control. The approximate 50% reduction in the current amplitude in the presence of these compounds or the prominent change in the current kinetics were an indicator of both the ion channel and the proper operation of the perfusion system.

We tested the effect of 0.2% DMSO, the maximum concentration that was applied to the cells at the highest ClGBI concentration, as it may affect not only the conductance of the channels but also the viability of cells [55,56,57]. We did not observe any changes either in the peak amplitude or the kinetics of the current when the cells were perfused with an ECS containing 0.2% DMSO (data not shown). Similar observations were found by others when the effect of DMSO was tested on different ion channels [58,59].

### 4.5. Voltage Protocols

In general, the holding potential (V_h_) was –120 mV, and the depolarizing pulses were delivered every 15 s, except when indicated. Depolarizing pulses to +50 mV ranging from 15 to 1500 ms were applied to record the currents of the K_V_1.1, K_V_1.3, fast inactivation-removed hK_V_1.4 (K_V_1.4ΔN), K_V_1.5, and K_V_10.1 channels. K_V_11.1 currents were evoked by a voltage step to +20 mV for 1.25 s from a V_h_ of –80 mV followed by a step to –40 mV for 2 s every 30 s, and the peak (tail) currents were recorded during the latter step. For K_Ca_3.1 currents, a 200-ms-long voltage ramp to +50 mV from –120 mV was applied every 10 s. For hK_Ca_3.1, the reversal potential for K^+^ was determined, and only those currents were analyzed for which the reversal potential fell into the range of the theoretical reversal potential ± 5 mV (–86.5 ± 5 mV). The current through the human proton channel (hH_V_1) was elicited by applying a 1.0-s-long voltage ramp to +60 mV from a V_h_ of –60 mV every 10 s. For sodium currents through Na_V_1.4 and Na_V_1.5, 10-ms-long voltage steps to 0 mV were applied every 10 s.

### 4.6. Patch Clamp Data Analysis

The pClamp 10.7 software package (Molecular Devices Inc., Sunnyvale, CA, USA) and GraphPad Prism 8 (Graphpad, CA, USA) were used for data acquisition and analysis. In general, currents were lowpass-filtered using the built-in analog four-pole Bessel filters of the amplifiers and were sampled (2–50 kHz) at least twice the filter cut-off frequency. Before analysis, current traces were digitally filtered with a 3-point boxcar filter and were corrected for ohmic leakage when needed.

The H_V_1 current recordings were evaluated as follows. First, the traces were filtered (lowpass boxcar, 3 smoothing points), and off-line leaks were corrected. As leaks are an ohmic current (i.e., the voltage–current relationship is linear), we defined a region where the opening probability of the H_V_1 channels is approximately zero. Thus, a linear regression line was fit to the data points between 16 ms to 80 ms, corresponding to −60 mV and −53 mV, and the fitted parameters were used to subtract non-specific leaks. The leak-corrected currents between +59 mV and +60 mV were extracted, averaged, and considered the peak current. The average currents of two or three stable traces at a given pH_ec_ condition defined one data point.

The inhibitory effect of ClGBI at a given concentration was calculated as the remaining current fraction (*RCF* = I/I_0_, where I_0_ is the peak current in the absence of ClGBI, and I is the peak current at equilibrium block at a given concentration of ClGBI). The data points (average of 3–5 individual records) in the dose–response curve was fitted with a two-parameter inhibitor vs. response model using
$$RCF=\frac{K_{d}^{n_{H}}}{K_{d}^{n_{H}}+{\left[ClGBI\right]}^{n_{H}}}$$
where [*ClGBI*] is the molar concentration of ClGBI, *K_d_* is the dissociation constant, and *n_H_* is the Hill coefficient. All data are presented as means ± SEM.

To examine the binding kinetics, *RCF* was plotted as a function of time. The association time constant (*τ_on_*) was determined by fitting the data points with a single exponential function,
$$RCF=RCF_{0}\times e^{-t/\tau_{on}}+C$$
where *RCF*_0_ is the *RCF* value before the addition of the drug, and *C* is a constant term.

## Figures and Tables

**Figure 1 pharmaceuticals-16-00656-f001:**
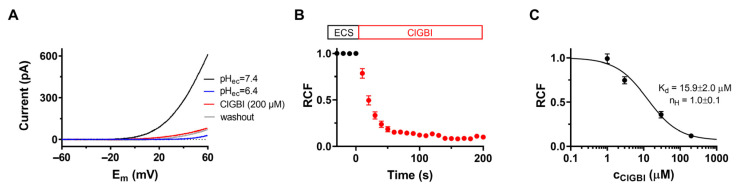
Inhibition of hH_V_1 currents by ClGBI. Peak currents were determined using protocols and experimental conditions as described in Materials and Methods. (**A**) Whole-cell currents were recorded on CHO cells expressing H_V_1 in response to slow voltage ramps (0.12 mV/ms) from −60 to +60 mV in the absence (pH_ec_ = 7.4, black) and presence of 200 μM ClGBI (ClGBI, red). ECS at a pH of 6.4 was used as an indicator of both the ion channel and the proper operation of the perfusion system (pH_ec_ = 6.4, blue). (**B**) The effect of ClGBI at 200 μM was determined as remaining current fraction (*RCF*, see Materials and Methods) and plotted as a function of time as 200 μM ClGBI was applied to the bath solution. Pulses were delivered every 10 s. (**C**) Concentration-dependent inhibition of the H_V_1 channel by ClGBI. Points on the dose–response curves represent the mean of 4–5 independent measurements. Data points were fitted with a two-parameter Hill equation (see Materials and Methods). The best fit yielded *K_d_* of 15.9 ± 2.0 μM and *n_H_* of 1.0 ± 0.1. Error bars represent SEM.

**Figure 2 pharmaceuticals-16-00656-f002:**
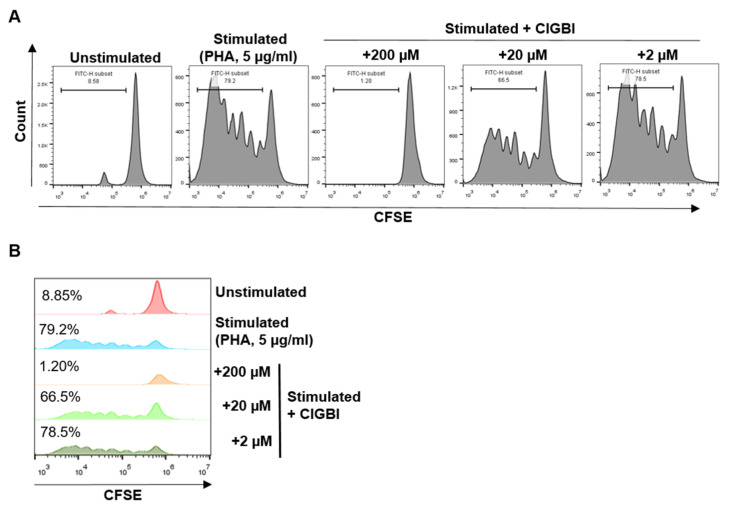
The effect of ClGBI on human lymphocyte proliferation. (**A**) Representative flow cytometry histograms of CFSE-labeled lymphocytes in the absence and presence of ClGBI (200 μM, 20 μM, and 2 μM). (**B**) Histogram overlays of samples presented in (**A**) showing the percentage of proliferating cells by CFSE dilution.

**Figure 3 pharmaceuticals-16-00656-f003:**
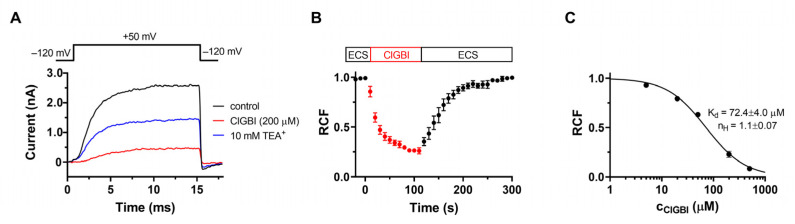
Block of K_V_1.3 channels by ClGBI. (**A**) Whole-cell potassium currents through hK_V_1.3 channels evoked from a human T cell by depolarizing pulses to +50 mV every 15 s. Traces indicate K^+^ current in the absence (control, black) and presence of 200 µM ClGBI at equilibrium block (ClGBI, red), with 10 mM TEA^+^ (blue) used as a positive control. (**B**) Kinetics of the block. Normalized peak currents as remaining current fraction (*RCF*) are plotted as a function of time. The boxes indicate the application of the control ECS and ClGBI-containing ECS, respectively. Solid line indicates the fitting of a single-exponential decaying function to the data points. Perfusion with ClGBI-free ECS resulted in complete recovery from block. Data points were recorded every 15 s. (**C**) Dose–response relationship for ClGBI on K_V_1.3. The dose response was obtained by plotting the remaining current fraction as a function of ClGBI concentration and fitting it with a two-parameter Hill equation (see Materials and Methods). Error bars indicate SEM throughout the figure.

**Figure 4 pharmaceuticals-16-00656-f004:**
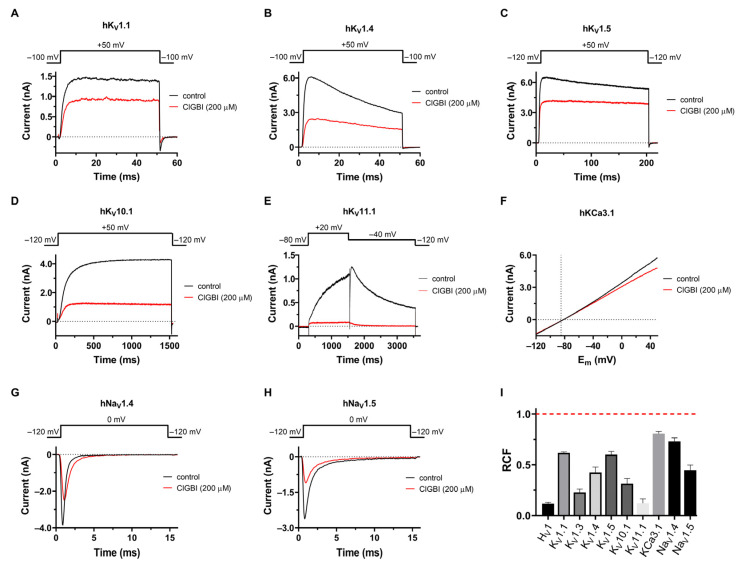
Selectivity profile of ClGBI. Effect of 200 μM ClGBI was assayed in CHO cells transfected with different ion channel genes or using HEK293 cell line stably expressing hK_V_11.1 and hNa_V_1.4 channels. The representative current traces shown here were measured before the application of ClGBI (control, black) and after reaching equilibrium block upon application of 200 μM ClGBI (red). (**A**–**D**) Currents were evoked from holding potentials of –100 mV by depolarizations to +50 mV for the time interval indicated on the panels. The time between depolarizing pulses was 15 s or 30 s (for K_V_10.1). (**E**) hK_V_11.1 currents were measured with a voltage step from a holding potential of –80 mV to +20 mV, followed by a step to –40 mV, during which the peak current was measured. Pulses were delivered every 30 s. (**F**) hK_Ca_3.1 currents were elicited every 15 s with voltage ramps to +50 mV from a holding potential of −120 mV at a rate of 0.85 mV/ms. (**G**,**H**) hNa_V_1.4 and hNa_V_1.5 currents were measured by applying depolarizing pulses to 0 mV from a holding potential of –120 mV every 15 s. (**I**) The effect of 200 μM ClGBI on the peak currents was reported as the remaining current fraction (*RCF* = I/I_0_, where I_0_ and I are the peak currents in the absence and presence of 200 μM ClGBI at equilibrium block, respectively). Red dashed line indicates no effect. Bars represent the mean of 3–8 independent measurements; error bars indicate the SEM.

## Data Availability

Data is available within the article.

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
