# Peer review of "5-Chloro-2-Guanidinobenzimidazole (ClGBI) Is a Non-Selective Inhibitor of the Human HV1 Channel"

_pharmaceuticals, 2023, doi:10.3390/ph16050656_

Round 1

Reviewer 1 Report

Comments to the Author

Summary

The manuscript with the title “5-chloro-2-guanidinobenzimidazole (C1GBI) is a non-selective inhibitor of the human HV1 channel” by Szanto et al, is well written and technically sound. The authors investigate a newly found inhibitor of voltage-gated proton channels. However, they find that the same substance has additional inhibiting effects on potassium and sodium channels. The results are very intriguing and of broader interest. I critique the way voltage-gated proton channels were investigated, as it is almost impossible to determine a maximal current using a voltage ramp protocol. However, one might accept that C1GBI as been tested to the utmost by other groups and its function as an inhibitor to HV1 is established. Still, the authors compare the Kd values with other channel species. One would doubt the numbers for HV1 are correct. Additionally, even if the authors mention it in the main text, it is not reasonable to determine a Kd value from a single concentration tested. Therefore, the conclusions presented in the discussion are not well-reinforced. Nonetheless, C1GBI appears to inhibit all channels tested in this work that have a classical VSD. If the authors can convincingly exclude any side effects of C1GBI than the results need to be published.

I mentioned some major and minor points the authors should address.

Major points:

1.      The authors measuring time constants of  ion channel inhibition. How fast is the solution exchange? The value of perfusion rate is given as 200 mL/min (line 448) but at which time is the solution completely exchanged in the chamber?

2.      The analysis of inhibition compares peak currents with or without inhibitor. How are these peaks determined? Especially regarding to proton currents this way of analysis appears not very precise. The authors should provide typical raw data and show the analysis in an example.   

3.      The authors should exclude that 5-chloro-2-guanidinobenzimidazole is not changing the pH in the cells or the media they use. In this case they would have measured pH sensitivity of the channels.

Minor points

1.      Several times the concentration of C1GBI is determined in mM (line 90, line 111, line 114, line 118, line 151, line 156, line 210, line 218, line 266, line 335).

2.      The CHO have been reported to express endogenously voltage-gated proton channels, are the authors able to show control recordings? [1]

3.      Figure legend 1A+1B. Is it 200 mM C1GBI or 200 µM C1GBI?

4.      alsorelies line 141 should be separated

5.      DeCoursey et al., 1984 should be cited ---------- DeCoursey TE, Chandy KG, Gupta S, Cahalan MD. Voltage-gated K+ channels in human T lymphocytes: a role in mitogenesis? Nature. 1984 Feb 2-8;307(5950):465-8. doi: 10.1038/307465a0. PMID: 6320007.

6.      Please cite Musset et al 2009 line 226------ Musset, B., Morgan, D., Cherny, V. V., MacGlashan, D. W., Jr., Thomas, L. L., Rios, E. & DeCoursey, T. E. (2008) A pH-stabilizing role of voltage-gated proton channels in IgE-mediated activation of human basophils, Proc Natl Acad Sci U S A. 105, 11020-5. doi:10.1073/pnas.0800886105

7.      Line 246 citation 45 appears misplaced after presenting CHO cells

8.      Why has the control solution for NaV channels a lower pH than the other used solutions?

9.      The authors use PBMCs. How did the authors avoid t patch monocytes and B cells?

References

10.            1.  Cherny, V. V., Henderson, L. M. & DeCoursey, T. E. (1997) Proton and chloride currents in Chinese hamster ovary cells, Membr Cell Biol. 11, 337-47.

Author Response

The manuscript with the title “5-chloro-2-guanidinobenzimidazole (C1GBI) is a non-selective inhibitor of the human HV1 channel” by Szanto et al, is well written and technically sound. The authors investigate a newly found inhibitor of voltage-gated proton channels. However, they find that the same substance has additional inhibiting effects on potassium and sodium channels. The results are very intriguing and of broader interest. I critique the way voltage-gated proton channels were investigated, as it is almost impossible to determine a maximal current using a voltage ramp protocol. However, one might accept that C1GBI as been tested to the utmost by other groups and its function as an inhibitor to HV1 is established. Still, the authors compare the Kd values with other channel species. One would doubt the numbers for HV1 are correct. Additionally, even if the authors mention it in the main text, it is not reasonable to determine a Kd value from a single concentration tested. Therefore, the conclusions presented in the discussion are not well-reinforced. Nonetheless, C1GBI appears to inhibit all channels tested in this work that have a classical VSD. If the authors can convincingly exclude any side effects of C1GBI than the results need to be published.

We would like to thank the reviewer for the critical comments and questions meant to improve the manuscript. We address all the raised issues below:

“it is almost impossible to determine a maximal current using a voltage ramp protocol.”

The ramp protocol obviously supplies different information from the step protocol as the current is measured as a function of both changing voltage and time. We prefer to use this protocol since it readily shows if the voltage-dependence of channel opening is changed, which happens for example when the transmembrane pH gradient is changed. When voltage-dependence and kinetics are not affected, the current measured at the end of the ramp at +60 mV remains stable and is a good indicator of the number of conducting channels. Due to the slow activation kinetics of Hv1, the current amplitude does not even saturate by the end of 1-second-long depolarizing steps so the actual value of the maximal current will depend on the length of the pulse in this case as well. Thus, we believe that conceptually the use of ramps is not an inferior technique to using steps when determining the amplitude.

“Still, the authors compare the Kd values with other channel species. One would doubt the numbers for HV1 are correct.”

Naturally, channels from different species may show different affinities for an inhibitor, although if the binding site is conserved, affinities are usually very similar. This is why all the channels we tested with ClGBI were all human so that the comparison would be more relevant. It is also reassuring that our Kd value on Hv1 in the mammalian expression system (16 mM) was very close to the Kd published for hHv1 in the Xenopus oocyte expression system (26 mM).

“Additionally, even if the authors mention it in the main text, it is not reasonable to determine a Kd value from a single concentration tested.”

We are aware that estimating Kd from a single concentration effect may be unreliable, but as we explain in the text, channels blocked with low affinity would have required very high ClGBI and also DMSO concentrations to complete dose-response curves. In our experience DMSO concentrations over 1% can cause prompt changes to currents and to the membrane, so such conditions should be avoided. Nevertheless, we believe that despite the inherent accuracy of the single-point estimate, the order of magnitude of the estimated Kd is correct and can be used for a comparative affinity sequence.

I mentioned some major and minor points the authors should address.

Major points:

  1. The authors measuring time constants of ion channel inhibition. How fast is the solution exchange? The value of perfusion rate is given as 200 mL/min (line 448) but at which time is the solution completely exchanged in the chamber?

The perfusion outlet is positioned close to and aimed directly at the measured cell, therefore the solution is exchanged around the cell within a few seconds of making the switch. In each case, we use a special solution to the test the effective operation of the system: for Hv1 we switch to pH 6.4 solution, which shifts the activation threshold to the right by about 40 mV and reduces the current, for K+ channels we use high K+ external solution to shift the reversal potential, etc. When the perfusion system is positioned correctly, the current will change by the next episode (usually applied every 10 or 15s) and remain stable; and then following the switch back to the control solution the current will recover within one episode as well. Unfortunately, during the conversion step of the manuscript upload process many (but interestingly not all) of our m (Greek mu) characters were changed to “m” in the text confusing the micro and milli prefixes. We checked our original Word file and it contains m correctly at all locations. The flow rate was actually 200 ml/min (microliter/min). However, the time required for complete solution change in the chamber is not relevant due to the direct fluid flow on the cell confirmed by the procedure described above. Moreover, for clarity, we have extended the representative current traces with the positive control experiments shown in Fig.1A (pH shift) and Fig3A (10 mM TEA+, that is the IC50 for KV1.3).

  1. The analysis of inhibition compares peak currents with or without inhibitor. How are these peaks determined? Especially regarding to proton currents this way of analysis appears not very precise. The authors should provide typical raw data and show the analysis in an example.

The Figures above show a representative peak HV1 current analysis. Top left corner shows a representative raw current data. First, the traces were filtered (lowpass boxcar, 3 smoothing points), and off-line leak corrected. As the leak is an ohmic current (i.e., the voltage-current relationship is linear) we can select a region, where the opening probability of the HV1 channels is approximately zero. Thus, a linear regression line was fit to the data points between 16 ms to 80 ms, corresponding to -60 mV and -53 mV and the fitted parameters were used to subtract the non-specific leak. At the bottom, an uncorrected (left), and a leak corrected HV1 current trace is shown (right). The leak-corrected currents between +59 mV and +60 mV were extracted, averaged and considered as the peak current. The average currents of two or three stable traces at a given pHext condition defined one data point.

  1. The authors should exclude that 5-chloro-2-guanidinobenzimidazole is not changing the pH in the cells or the media they use. In this case they would have measured pH sensitivity of the channels.

We have checked the pH of the control solution for HV1 measurement and compared the results with 200 µM ClGBI dissolved in HV1 solution. The difference in pH between the two solutions was less than 0.05.

Minor points

  1. Several times the concentration of C1GBI is determined in mM (line 90, line 111, line 114, line 118, line 151, line 156, line 210, line 218, line 266, line 335).

We apologize for this, but unfortunately, during the conversion step of the manuscript upload process many (but interestingly not all) of our m (Greek mu) characters were changed to “m” in the text confusing the micro and milli prefixes. We checked our original Word file and it contains m correctly at all locations. We have corrected them and hope it does not happen again. We have checked the uploaded file, as well and found no further mistakes.

  1. The CHO have been reported to express endogenously voltage-gated proton channels, are the authors able to show control recordings?

Thank you for the comment. Although it has been reported that voltage-gated proton channels are present on the membrane of native CHO cells, in our system we could record £10 pA currents at +60 mV, on average (see a representative raw data at the bottom). This endogenous HV1 currents are negligible in transfected cells, and thus, do not affect the ClGBI inhibition at 200 micromolar concentration.

  1. Figure legend 1A+1B. Is it 200 mM C1GBI or 200 µM C1GBI?

Thank you for the comment. It was 200 mM, we have corrected in the main text.

  1. alsorelies line 141 should be separated

Thank you for the comment. We have modified it in the main text.

  1. DeCoursey et al., 1984 should be cited ---------- DeCoursey TE, Chandy KG, Gupta S, Cahalan MD. Voltage-gated K+ channels in human T lymphocytes: a role in mitogenesis? Nature. 1984 Feb 2-8;307(5950):465-8. doi: 10.1038/307465a0. PMID: 6320007.

We have added this citation and refreshed the references.

  1. Please cite Musset et al 2009 line 226------ Musset, B., Morgan, D., Cherny, V. V., MacGlashan, D. W., Jr., Thomas, L. L., Rios, E. & DeCoursey, T. E. (2008) A pH-stabilizing role of voltage-gated proton channels in IgE-mediated activation of human basophils, Proc Natl Acad Sci U S A. 105, 11020-5. doi:10.1073/pnas.0800886105

We have added this citation and refreshed the references.

  1. Line 246 citation 45 appears misplaced after presenting CHO cells

Thank you for the comment. The original citation was related to HEK cells. We have removed this citation from the text.

  1. Why has the control solution for NaV channels a lower pH than the other used solutions?

For NaV current recordings the pH of the ECS was 7.35, similar to the recordings of KV currents. For KV recordings we used a KF based solution whose pH was 7.2. For NaV channels, although a Cs+-based ICS was used but its pH was also 7.2. We have double checked the main text and found no mistake. As a matter of fact, the pH of ICS was 7.2 almost for all channels, except KV11.1, for which it was 7.3. But this quite slight difference in pH has no effect on the current inhibition.

  1. The authors use PBMCs. How did the authors avoid t patch monocytes and B cells?

The majority of PBMCs are T cells so statistically they would be patched with the highest probability. Monocytes can be avoided based on their significantly larger size than lymphocytes. Although we cannot distinguish B cells from T cells, the main potassium channel of B cells is also Kv1.3, which is the dominant current under the conditions we used for patching. Therefore, our conclusion regarding the effect of ClGBI on Kv1.3 would not be affected by accidental patching of a few B cells.

References

  1. 1. Cherny, V. V., Henderson, L. M. & DeCoursey, T. E. (1997) Proton and chloride currents in Chinese hamster ovary cells, Membr Cell Biol. 11, 337-47.

We have inserted this citation in the main text, thank you for sharing this publication with us.

Reviewer 2 Report

In this work, the authors examined the ion channel selectivity of a known and widely used molecule, 5-chloro-2-guanidinobenzimidazole (ClGBI), usually recognized to specifically target the proton channel Hv1 with a Kd of approximately 26 µM. However, a comprehensive study of its ion channel selectivity has not been published yet. In this study, the authors found that ClGBI inhibits the proliferation of lymphocytes, which absolutely requires the activity of the Kv1.3 channel. Therefore, they tested ClGBI directly on hKv1.3 using whole-cell patch-clamp recording in CHO cells and found an inhibitory effect similar in magnitude to that seen on hV1 (Kd of ≈72 µM). They further investigated ClGBI selectivity on hKV1.1, hKV1.4-IR, hKV1.5, hKV10.1, hKV11.1, hKCa3.1, hNaV1.4 and hNaV1.5 channels and their results show that except hKCa3.1, ClGBI exhibits a significant inhibition of these ion channels. Base on their results, the authors conclude that ClGBI has to be considered as a non-selective hHV1 inhibitor; thus experiments aiming at elucidating the significance of these channels in physiological responses have to be carefully evaluated.

This study is interesting and has significant relevance because the results provide helpful information on the lack of selectivity of a drug widely used to target specifically Hv1 proton channels. The results delivered by the authors provide evidence that one must be very careful when using ClGBI and drawing conclusions about its putative function. The experiments are carefully performed and the results are clearly presented. Nonetheless, I have few concerns, which need to be addressed by the authors:

1- All along the manuscript, the concentrations of ClGBI are a mess of either mM or µM expression. Please, correct it in a consistent fashion.

2- In the discussion, the authors conclude that ClGBI likely act at pore of hKV1.1, hKV1.4-IR, hKV1.5, hKV10.1, hKV11.1, hNaV1.4 and hNaV1.5 channels and not at their VSD. I would be more careful in drawing such a conclusion and the authors should tone down this assumption.

Author Response

First of all, we would like to thank Reviewer 2 for the time and efforts devoted to reviewing our manuscript. We appreciate the comments as well, which were addressed as follows:

In this work, the authors examined the ion channel selectivity of a known and widely used molecule, 5-chloro-2-guanidinobenzimidazole (ClGBI), usually recognized to specifically target the proton channel Hv1 with a Kd of approximately 26 µM. However, a comprehensive study of its ion channel selectivity has not been published yet. In this study, the authors found that ClGBI inhibits the proliferation of lymphocytes, which absolutely requires the activity of the Kv1.3 channel. Therefore, they tested ClGBI directly on hKv1.3 using whole-cell patch-clamp recording in CHO cells and found an inhibitory effect similar in magnitude to that seen on hV1 (Kd of ≈72 µM). They further investigated ClGBI selectivity on hKV1.1, hKV1.4-IR, hKV1.5, hKV10.1, hKV11.1, hKCa3.1, hNaV1.4 and hNaV1.5 channels and their results show that except hKCa3.1, ClGBI exhibits a significant inhibition of these ion channels. Base on their results, the authors conclude that ClGBI has to be considered as a non-selective hHV1 inhibitor; thus experiments aiming at elucidating the significance of these channels in physiological responses have to be carefully evaluated.

This study is interesting and has significant relevance because the results provide helpful information on the lack of selectivity of a drug widely used to target specifically Hv1 proton channels. The results delivered by the authors provide evidence that one must be very careful when using ClGBI and drawing conclusions about its putative function. The experiments are carefully performed and the results are clearly presented. Nonetheless, I have few concerns, which need to be addressed by the authors:

We would like to thank the reviewer for the time and effort devoted to the review of our manuscript and the overall positive opinion. We address the raised concerns below.

1- All along the manuscript, the concentrations of ClGBI are a mess of either mM or µM expression. Please, correct it in a consistent fashion.

We apologize for this, but unfortunately, during the conversion step of the manuscript upload process many (but interestingly not all) of our m (Greek mu) characters were changed to “m” in the text confusing the micro and milli prefixes. We checked our original Word file and it contains m correctly at all locations. We have corrected them and hope it does not happen again. We have checked the uploaded file, as well and found no further mistakes.

2- In the discussion, the authors conclude that ClGBI likely act at pore of hKV1.1, hKV1.4-IR, hKV1.5, hKV10.1, hKV11.1, hNaV1.4 and hNaV1.5 channels and not at their VSD. I would be more careful in drawing such a conclusion and the authors should tone down this assumption.

We agree that our conclusion was far-fetched so we rephrased that paragraph to state a more conservative conclusion.

Round 2

Reviewer 1 Report

All points I have raised have been addressed. I see no reason why the manuscript should not be published.